# IMPROVING GENERALIZATION FOR SMALL DATASETS WITH DATA-AWARE DYNAMIC REINITIALIZATION

## ABSTRACT

The efficacy of deep learning techniques is contingent upon copious volumes of data (labeled or unlabeled). Nevertheless, access to such data is frequently restricted in practical domains such as medical applications. This presents a formidable obstacle: How can we effectively train a deep neural network on a relatively small dataset while improving generalization? Recent works explored evolutionary or iterative training paradigms, which reinitialize a subset of the parameters to improve generalization performance for small datasets. While effective, these methods randomly select the subset of parameters and maintain a fixed mask throughout iterative training, which can be suboptimal. Motivated by the process of neurogenesis in the brain, we propose a novel iterative training framework, Selective Knowledge Evolution (SKE), that employs a data-aware dynamic masking scheme to eliminate redundant connections by estimating their significance, thereby increasing the model's capacity for further learning via random weight reinitialization. The experimental results demonstrate that our approach outperforms existing methods in accuracy and robustness, highlighting its potential for real-world applications where collecting data is challenging. [1]

## 1 INTRODUCTION

Deep neural networks (DNNs) have become essential for solving complex problems in various fields, such as image and speech recognition, natural language processing, and robotics (LeCun et al., 2015). With the increasing availability of data, DNNs have achieved unprecedented performance, surpassing human-level performance in some applications (Silver et al., 2016). However, the success of DNNs is limited when dealing with small datasets, where the model tends to overfit and fails to generalize to new data. For example, it is often difficult to obtain a large amount of data in medical diagnosis due to the complexity and high cost of the procedures involved. In such cases, lack of generalization can be dangerous, which can lead to incorrect diagnosis and treatment.

Recently, several studies based on weight reinitialization methods (Han et al., 2016; Furlanello et al., 2018) have been proposed in the literature to improve generalization by iteratively refining the learned solution through partial weight reinitialization. These methods select and retain a subset of parameters while randomly reinitializing the rest of the network during iterative/evolutionary training schemes. For example, a state-of-the-art method named Knowledge Evolution (KE) (Taha et al., 2021) improves generalization by randomly splitting the network into fit and reset subnetworks and constantly reinitializing the reset subnetwork after each iteration. However, the KE approach is limited by its reliance on a predetermined mask creation, where a random subset of parameters is selected and kept constant throughout the iterative training process. This constraint may impede the model's ability to learn effectively from small datasets, ultimately limiting its generalization capabilities. These limitations raise two important questions: 1) Can we leverage an evolutionary training paradigm to evolve or adapt the mask over generations, instead of using a fixed mask, in order to enhance the generalization performance of deep learning models trained on small datasets? 2) Can we utilize the available data and the internal state of the model to dynamically determine the important parameters for each generation, rather than randomly presetting them?

In our quest to address these questions, we draw inspiration from the phenomenon of neurogenesis in the brain. Neurogenesis, the process of dynamically generating or eliminating neurons in

---

[1]Upon acceptance, the source code will be made available.

response to environmental demands, has been found to play a crucial role in learning and memory consolidation (Shors et al., 2001; Garthe et al., 2016; Kempermann et al., 2015). This intricate process enables the brain to adapt to new experiences and stimuli, enhancing generalizability. Recent advances in neuroscience have shed light on the non-random integration of new neurons within the brain (Yasuda et al., 2011). For instance, in rodents, neurogenesis-dependent refinement of synaptic connections has been observed in the hippocampus, where the integration of new neurons leads to the elimination of less active synaptic connections (Aimone et al., 2014; Vadodaria & Gage, 2014). Selective neurogenesis is critical to improving generalization ability by providing a diverse pool of neurons with distinct properties that can integrate into existing neural networks and contribute to adaptive learning (Toni et al., 2008). Although the precise mechanisms that govern selective neurogenesis are not fully understood, these findings suggest that selective neurogenesis in the human brain enhances generalization capabilities through its dynamic and selective nature. Thus, by emulating the characteristics of selective neurogenesis, we unlock its potential to improve generalization in deep neural networks.

Therefore, we present a novel iterative training approach called Selective Knowledge Evolution (SKE), which distinguishes itself from the conventional Knowledge Evolution (KE) method through its mask computation. Unlike a predetermined fixed mask, SKE utilizes a data-aware dynamic masking criterion that evolves and adapts the mask over generations. Through extensive experiments on multiple datasets, we demonstrate that our proposed training paradigm greatly improves the performance and generalization of the models. Furthermore, SKE effectively addresses overfitting on relatively small datasets, alleviating the need for extensive data collection.

The main contributions of the paper are as follows.

- Selective Knowledge Evolution (SKE) is an evolutionary training paradigm that incorporates data-aware dynamic masking to selectively transfer knowledge across generations.

- Our proposed training paradigm facilitates the learning of generalizable features and increases the overall performance of DNNs across small datasets.

- SKE exhibits robustness in solving more common challenges in real-world problems, including learning with class imbalance, natural corruption, and adversarial attacks.

## 2 RELATED WORK

Iterative training and weight reinitialization for DNNs is a prominent area of research (Taha et al., 2021; Furlanello et al., 2018; Oh et al., 2022; Zaidi et al., 2023) that focuses mainly on improving generalization performance by partially refining the learned solution or fully iterating the learned solution. Dense-Sparse-Dense (DSD) (Han et al., 2016) propose a three-phase approach where weights with small magnitudes are pruned after initial training to induce sparsity and retrain the network by reinitializing the pruned weights to zero. Oh et al. (2022) proposed a weight reinitialization method to enhance cross-domain few-shot learning performance. Zaidi et al. (2022)conducted an extensive investigation into the conditions under which reinitialization proves beneficial. BANs (Born Again Neural Networks) (Furlanello et al., 2018) is a knowledge-distillation-based method that follows a similar iterative training paradigm. However, the critical difference between our work and BANs is that it employs the class-logits distribution instead of the network weights to transfer knowledge between successive networks. Recently, Knowledge Evolution (KE) (Taha et al., 2021) splits model weights into fit and reset parts randomly and iteratively reinitializes the reset part during training. The splitting method can be arbitrary (weight-level splitting (WELS)) or structured (Kernel-level convolutional-aware splitting (KELS)). This approach involves perturbing the reset hypothesis to evolve the knowledge within the fit hypothesis over multiple generations. Our framework (SKE) distinguishes itself from the conventional Knowledge Evolution (KE) method through its mask computation. SKE utilizes data-aware dynamic masking that adapts the mask over generations and transfers selective knowledge.

Additionally, we distance our work from the existing literature on neural architecture search (NAS) (Gordon et al., 2018) and growing neural networks (Evci et al., 2022). Specifically, we focus on a fixed network architecture, assuming that the connections and parameter count remain constant throughout our analysis. Finally, our work distinguishes itself from the dynamic sparse training

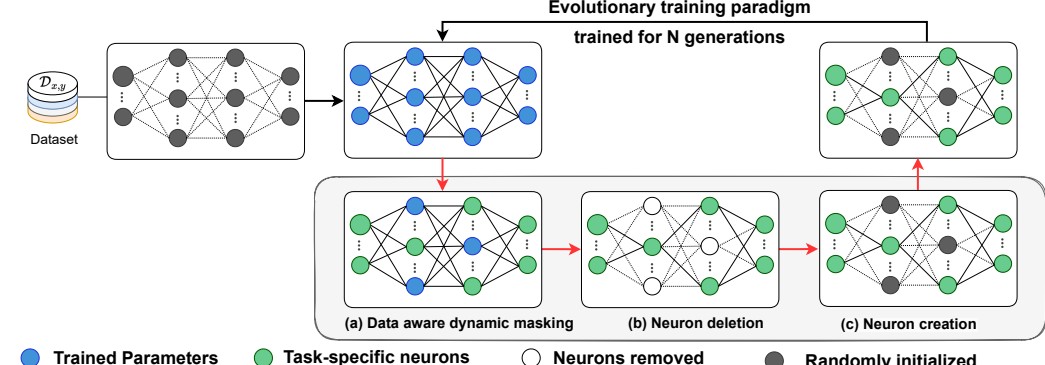

Figure 1: Schematics of proposed *Selective Knowledge Evolution (SKE)* framework. Our framework utilizes a data-aware dynamic masking scheme to remove redundant connections and increase the network's capacity for further learning by incorporating random weight reinitialization. Thus, effectively improving the performance and generalization of deep neural networks on small datasets.

literature (Evci et al., 2020; Liu et al., 2021) as our goal is not to acheive sparsity rather to improve generalization on small datasets.

## 3 METHODOLOGY

### 3.1 EVOLUTIONARY TRAINING PARADIGM

We first introduce the evolutionary/iterative training paradigm as envisioned in KE (Taha et al., 2021). Evolutionary training paradigms allow neural networks to be trained for many generations, where each generation focuses on optimizing the model to converge towards a local minimum while progressively improving generalization. Each generation within the training process is denoted as $g$, where $g$ ranges from 1 to the total number of generations, $N$.

We define a deep neural network $f$ with $L$ layers and is characterized by the set of parameters $\Theta$. We assume a dataset $D$ consisting of $n$ input-output pairs, denoted $\{(x_i, y_i)\}_{i=1}^{n}$. For a classification task, we define the cross entropy loss for training the network as:

$$\mathcal{L}_{ce} = -\frac{1}{n} \sum_{i=1}^{n} [y_i \log(\text{softmax}(f(x_i; \Theta))) + (1 - y_i) \log(1 - \text{softmax}(f(x_i; \Theta)))] \qquad (1)$$

where $\hat{y}_i = f(x_i)$ is the predicted output of the network for input $x_i$. We initialize the weights and biases of the network randomly.

KE starts by introducing a binary mask $M$, which partitions the weights of the neural network into two hypotheses before starting training: the fit hypothesis $H_{\text{fit}}$ and the reset hypothesis $H_{\text{reset}}$. This partitioning is expressed as follows:

$$H_{\text{fit}} = \mathbf{M} \odot \Theta \quad \text{and} \quad H_{\text{reset}} = (\mathbf{1} - \mathbf{M}) \odot \Theta \qquad (2)$$

Here, the element-wise multiplication operator $\odot$ is applied to the mask $M$ and the parameter set $\Theta$ to obtain the fit hypothesis $H_{\text{fit}}$. Similarly, the reset hypothesis $H_{\text{reset}}$ is obtained by element-wise multiplying the complement of the mask $(1 - M)$ with the parameter set $\Theta$. These parameters are chosen at random before the start of the first generation. This binary mask M is kept constant throughout the evolutionary training; i.e., the parameters belonging to the fit and reset hypotheses remain in that category across all generations.

We use the stochastic gradient descent (SGD) algorithm to train the network with a learning rate $\alpha$. We run SGD for $e$ epochs on the dataset $D$. The beginning of every new generation is characterized by introducing perturbations applied to the network weights to induce a high loss. This is done by reinitializing the parameters in the reset hypothesis while transferring or retaining the parameters belonging to the fit hypothesis. This dynamic process triggers a subsequent round of optimization,

guiding the neural network toward the search for a new minimum in the parameter space. The initialization of the network $f$ for the next generation $f_g$ is as follows:

$$\Theta_g \leftarrow \mathbf{M} \odot \Theta_{g-1} + (\mathbf{1} - \mathbf{M}) \odot \Theta_{Reinit} \tag{3}$$

where $\Theta_{g-1}$ and $\Theta_g$ are the parameters of the network $f$ belonging to the previous generation and current generation, respectively. $\Theta_{Reinit}$ corresponds to the randomly initialized tensor sampled from a uniform distribution. We then train the next generation of the network $f_g$ using SGD with the same hyperparameters and epochs as the first generation.

## 3.2 SELECTIVE KNOWLEDGE EVOLUTION (SKE) WITH DATA-AWARE DYNAMIC MASKING

Unlike KE, we propose a methodology that offers a distinct advantage regarding binary mask computation and parameter reinitialization. Motivated by the symbiotic link between generalization and selective neurogenesis in biological neural networks (Yasuda et al., 2011), we introduce a *Data-aware Dynamic Masking* (DDM) that emulates the process of selective neurogenesis in evolutionary training. The benefits of DDM's way of reinitialization are two-fold. 1) It takes advantage of the evolutionary training paradigm and adapts the mask dynamically in each generation rather than using a predetermined mask. This introduces flexibility in the network and improves the generalization performance of deep learning models trained on small datasets. 2) Our masking scheme leverages a model's data and internal state to dynamically determine the important parameters for a given task, rather than relying on random pre-setting, to enhance the performance of deep learning models on small datasets. Our way of masking offers a priori knowledge of where and what parameters and layers should be reinitialized in the general case.

The mask $M$ is calculated at the beginning of each generation in a data-dependent manner. We assess the importance or sensitivity of each connection in the network to the specific task by employing the SNIP method (Lee et al., 2018). SNIP decouples the connection weight from the loss function to identify relevant connections. We randomly sample a small subset of data ($\pi$) from the current dataset to evaluate connection sensitivity. We define a connection sensitivity mask $\mathbf{M} \in \{0,1\}^{|\Theta|}$, where $|\Theta|$ denotes the number of parameters in the network. The mask is designed to maintain a sparsity constraint $k$, which specifies the percentage of parameters to retain. The computation of connection sensitivity is performed as follows:

$$g_j(\mathbf{\Theta}; \pi) = \lim_{\delta \to 0} \frac{\mathcal{L}_{ce}(\mathbf{M} \odot \mathbf{\Theta}; \pi) - \mathcal{L}_{ce}((\mathbf{M} - \delta \mathbf{e}_j) \odot \mathbf{\Theta}; \pi)}{\delta} \bigg|_{\mathbf{M}=1} \tag{4}$$

where $j$ corresponds to the parameter index and $e_j$ is the mask vector of the index $j$, where the magnitude of the derivatives is then used to calculate the saliency criteria ($s_j$):

$$s_j = \frac{|g_j(\mathbf{\Theta}; \pi)|}{\sum_{k=1}^{m} |g_k(\mathbf{\Theta}; \pi)|}. \tag{5}$$

After calculating the saliency values, we apply the sparsity constraint $k$ to the connection sensitivity mask, which ensures that only the top-k task-specific connections are retained. The sparsity constraint $k$ is defined as follows:

$$\mathbf{M}_j = \mathbb{1}\left[s_j - \tilde{s}_\kappa \geq 0\right], \quad \forall j \in \{1 \dots m\}, \tag{6}$$

where $\tilde{s}_k$ is the $k^{\text{th}}$ largest element in the saliency vector $s$ and $\mathbb{1}[.]$ is the indicator function. Subsequently, using the saliency values obtained from the connection sensitivity analysis, we select and preserve the top-k important connections. The parameters associated with the connections deemed less important for the current generation are then reinitialized. This process effectively induces selective neurogenesis, allowing the network to adapt and free up its capacity for learning more generalized representations in subsequent generations. Finally, the network for subsequent generation is initialized as shown in Equation 3.

Intuitively, we incorporated selective neurogenesis as a replacement mechanism, reinitializing the input and output synaptic weights of specific subsets of network parameters dynamically during the evolutionary training process (Tran et al., 2022). Due to the challenges associated with where, how, and when to create neurons (Evci et al., 2022), we explore data-aware dynamic masking to drive neuron creation and removal, which could improve learning. We first select the crucial parameters

based on the computed saliency mask. Ideally, we would like the mask to keep the knowledge learned from the previous generation as much as possible and to have enough learning capacity to accommodate the learning happening in the new generation. The additional learning capacity facilitates the fast adoption of generalized knowledge and reinforces the knowledge retained from the previous generation. In this way, selective neurogenesis is achieved that inherently adapts the network connectivity patterns in a data-dependent way to learn generalized representations without altering overall network size.

The network with the new initialization undergoes next-generation training with the same data for the $e$ epochs, where $e$ is kept the same for each generation. The network is trained with the loss function shown in Equation 1. Thus, we favor the preservation of the task-specific connections more precisely than the mask criteria used in KE that can guide the network towards those desirable traits that efficiently improve the performance and generalization of DNNs in small datasets.

## 4 EXPERIMENTS AND RESULTS

Here, we provide the details on the experimental setup, implementation details, and datasets used in our empirical evaluation.

**Datasets:** We evaluate the proposed method using five datasets: Flower102 (Nilsback & Zisserman, 2008), CUB-200-2011 (Wah et al., 2011), MIT64 (Quattoni & Torralba, 2009), Stanford Dogs (Khosla et al., 2011), FGVC-Aircraft (Maji et al., 2013). The summaries of the statistics of the data set are mentioned in Appendix.

**Implementation Details:** Since our framework is a direct extension of the KE, we follow the same experimental setup. The efficacy of our framework is demonstrated in two widely used architectures: ResNet18 and ResNet50 (He et al., 2016). We randomly initialize the networks and optimize them with stochastic gradient descent (SGD) with momentum 0.9 and weight decay $1e - 4$. We use the cosine learning rate decay with an initial learning rate lr = {0.1, 0.256} on specific datasets. The networks are trained iteratively for $N$ generations ($N$=11) with a batch size $b$=32 for $e$=200 epochs without early stopping. The standard data augmentation technique, such as flipping and random cropping, are used. We employ SNIP (Lee et al., 2018) with network sparsity $k$ to find the critical subset of parameters at the end of each generation. For the importance estimation, we use 20% of the whole dataset as a subset ($\pi$). For all our experiments, we reinitialize a fixed 20% parameters of the network globally. All training settings (lr, $b$, $e$) are constant throughout generations.

**Baselines:** To evaluate and benchmark the effectiveness of our proposed approach, we conduct a comprehensive evaluation by comparing it against several existing methods that involve iterative retraining and reinitialization. Specifically, we benchmark our method against the following techniques: 1) Dense-Sparse-Dense Networks (DSD) (Han et al., 2016); 2) Born Again Networks (BANs) (Furlanello et al., 2018); and 3) Knowledge Evolution (KE) (Taha et al., 2021) (KELS). We also compare our method against a non-iterative approach known as the Long Baseline (LB), which undergoes training for the same number of epochs as the corresponding iterative methods. Since our framework is built on top of KE, we follow the same procedure in all our experiments unless specified.

### 4.1 CLASSIFICATION

Table 1 presents the quantitative classification evaluation results using ResNet18. $f_g$ denotes the result at the end of $g^{th}$ generation. We compare the performance at the end of $f_3$ (representing short-term benefits) and $f_{10}$ (representing long-term benefits) to assess the effectiveness of our approach. We compare SKE with two different configurations: (1) using naive cross-entropy loss (CE), and (2) incorporating label smoothing (Smth) regularizer with a hyperparameter $\alpha = 0.1$ (Müller et al., 2019).

The Selective Knowledge Evolution (SKE) framework demonstrates flexibility and consistently improves performance over the considered baselines across datasets. Interestingly, KE underperforms in terms of performance compared to long baseline (LB) with equal computation cost. This discrepancy may be attributed to the use of fixed masking criteria throughout evolutionary training, limiting

Table 1: Compares the results of our method with the other weight reinitialization methods on ResNet18. $g$ in $f_g$ indicates the number of generations the model is trained.

| Methods | Small Datasets | | | | |
|---|---|---|---|---|---|
| | CUB | Aircraft | Dog | Flower | MIT |
| CE ($f_1$) | 53.57 $_{\pm0.20}$ | 51.28 $_{\pm0.65}$ | 63.83$_{\pm0.12}$ | 48.48$_{\pm0.65}$ | 55.28$_{\pm0.19}$ |
| CE + DSD | 53.00$_{\pm0.32}$ | **57.24**$_{\pm0.21}$ | 63.58$_{\pm0.14}$ | 51.39$_{\pm0.19}$ | 53.21$_{\pm0.37}$ |
| CE + BAN ($f_{10}$) | 53.71$_{\pm0.35}$ | 53.19$_{\pm0.22}$ | 64.16$_{\pm0.13}$ | 48.53$_{\pm0.17}$ | 55.65$_{\pm0.28}$ |
| CE + KE ($f_{10}$) | 58.11$_{\pm0.25}$ | 53.21$_{\pm0.43}$ | 64.56$_{\pm0.31}$ | 56.15$_{\pm0.19}$ | 58.33$_{\pm0.43}$ |
| CE + SKE ($f_{10}$) | **59.72**$_{\pm0.21}$ | 55.87$_{\pm0.47}$ | **65.76**$_{\pm0.13}$ | **58.10**$_{\pm0.24}$ | **61.78**$_{\pm0.36}$ |
| Smth ($f_1$) | 58.92 $_{\pm0.24}$ | 57.16 $_{\pm0.91}$ | 63.64 $_{\pm0.16}$ | 51.02 $_{\pm0.09}$ | 57.74$_{\pm0.39}$ |
| Smth + LB ($f_3$) | 66.03$_{\pm0.13}$ | 62.55$_{\pm0.25}$ | 65.39$_{\pm0.55}$ | 59.51$_{\pm0.17}$ | 59.53$_{\pm0.60}$ |
| Smth + KE ($f_3$) | 62.88$_{\pm0.39}$ | 60.56$_{\pm0.36}$ | 64.23$_{\pm0.05}$ | 56.87$_{\pm0.65}$ | 58.78$_{\pm0.54}$ |
| Smth + SKE ($f_3$) | **68.56**$_{\pm0.24}$ | **64.37**$_{\pm0.19}$ | **65.72**$_{\pm0.15}$ | **62.13**$_{\pm0.23}$ | **62.62**$_{\pm0.51}$ |
| Smth + LB ($f_{10}$) | 69.80$_{\pm0.13}$ | 65.29$_{\pm0.51}$ | 66.19$_{\pm0.03}$ | 66.89$_{\pm0.23}$ | 61.29$_{\pm0.49}$ |
| Smth + KE ($f_{10}$) | 66.51$_{\pm0.070}$ | 63.32$_{\pm0.30}$ | 63.86$_{\pm0.21}$ | 62.56$_{\pm0.17}$ | 59.58$_{\pm0.62}$ |
| Smth + SKE ($f_{10}$) | **71.37**$_{\pm0.22}$ | **66.63**$_{\pm0.37}$ | **66.81**$_{\pm0.20}$ | **68.36**$_{\pm0.14}$ | **64.10**$_{\pm0.58}$ |

Table 2: Compares the results of the SKE framework with the KE and longer baselines for ResNet50 on large datasets. $g$ in $f_g$ indicates the number of generations the model is trained.

| Methods | Large datasets | | |
|---|---|---|---|
| | CIFAR10 | CIFAR100 | TinyImageNet |
| Smth ($f_1$) | 94.32 | 73.83 | 54.15 |
| Smth + LB ($f_{10}$) | 93.60 | 74.21 | 51.16 |
| Smth + KE ($f_{10}$) | 93.50 | 73.92 | 52.56 |
| Smth + SKE ($f_{10}$) | **94.61** | **75.05** | **54.50** |

the model's adaptability. In contrast, SKE outperforms both longer baselines and KE, consistently improving generalization performance across all datasets.

Similarly, we compare the performance of our method with the label smoothing regularizer (Smth) applied to the baselines. Table 1 shows that our method consistently outperforms the baselines with label smoothing on all datasets across generations. The combination of our selective knowledge evolution approach with label smoothing regularization leads to improved performance compared to using CE. These results demonstrate the efficacy of the data-aware dynamic masking and selective reinitialization employed by SKE. By adapting task-specific parameters in each generation, SKE achieves superior performance and enhances the model's generalization.

## 4.2 RESULTS ON LARGE DATASETS

Our work is a direct extension of KE (Taha et al., 2021), which focuses explicitly on improving generalization in the low data regime. However, we also thoroughly evaluate our method on large datasets such as Tiny-ImageNet (Le & Yang, 2015), CIFAR10, and CIFAR100 (Krizhevsky et al., 2009) using ResNet50 to assess its scalability. Table 2 compares the effectiveness of our method (SKE) with Knowledge Evolution (KE) and longer baseline (LB) in larger data sets. For each model, we trained it on top of the baseline for a specific number of generations ($f_{10}$), where N indicates the number of generations. The proposed approach exhibits promising performance and generalization across various large-scale datasets, such as TinyImageNet, when compared to KE and longer baselines. Furthermore, while the performance of KE and longer baselines (LB) falls below the normal standard training ($f_1$), the SKE framework demonstrates comparable or slightly improved performance in this scenario. This suggests that a selective way of reinitializing benefits iterative training and can effectively handle the complexities and challenges associated with larger datasets and architectures.

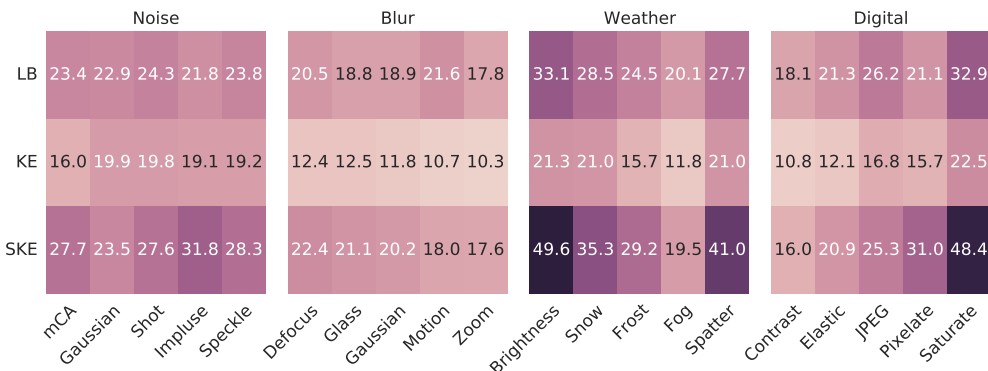

Figure 2: Robustness to natural corruptions on CIFAR10-C (Hendrycks & Dietterich, 2019). SKE is more robust against the majority of corruptions compared to the baselines.

## 5 ROBUSTNESS ANALYSES

### 5.1 ROBUSTNESS TO NATURAL CORRUPTIONS

In practical applications, deep neural networks often operate in dynamic environments characterized by variations such as lighting and weather conditions. Consequently, it is crucial to assess the robustness of DNNs to data distributions that undergo natural corruption. We investigate the robustness of DNNs to 15 common types of corruptions using the CIFAR-10-C dataset (Hendrycks & Dietterich, 2019). Our models are trained on clean images of CUB dataset and evaluated on CIFAR-10-C (Hendrycks & Dietterich, 2019). To quantify the performance under natural corruption, we use the Mean Corruption Accuracy (mCA) metric.

$$\text{mCA} = \frac{1}{N_c \times N_s} \sum_{c=1}^{N_c} \sum_{s=1}^{N_s} A_{c,s} \tag{7}$$

where $N_c$ and $N_s$ represent the number of corruptions (in this case, 19) and the number of severity levels (in this case, 5), respectively. Figure 2 illustrates the average accuracy of the models across 19 different corruptions at five severity levels. Notably, our proposed method (SKE) achieves a higher mCE (27.7%) compared to the longer baseline (23.4%) and KE (16%), demonstrating its effectiveness in improving the robustness to various types of corruption. These findings highlight the benefits of selectively reinitializing network parameters using a data-aware masking approach, resulting in enhanced robustness to natural corruptions.

### 5.2 ROBUSTNESS TO ADVERSARIAL ATTACKS

DNNs are vulnerable to adversarial attacks, where imperceptible perturbations are added to the input during inference to deceive the network and induce false predictions (Szegedy et al., 2013). Therefore, we investigate the robustness of DNNs trained against adversarial attacks using the PGD-10 attack (Madry et al., 2017) on models trained on the CIFAR10 dataset. We vary the intensity of the PGD attack and evaluate the models' performance. As shown in Figure 3(left), our proposed framework (SKE) exhibits greater resistance to adversarial attacks across different attack strengths compared to KE and the long

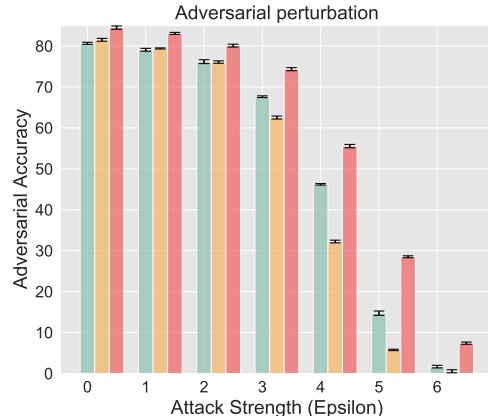

Figure 3: Robustness to adversarial attacks

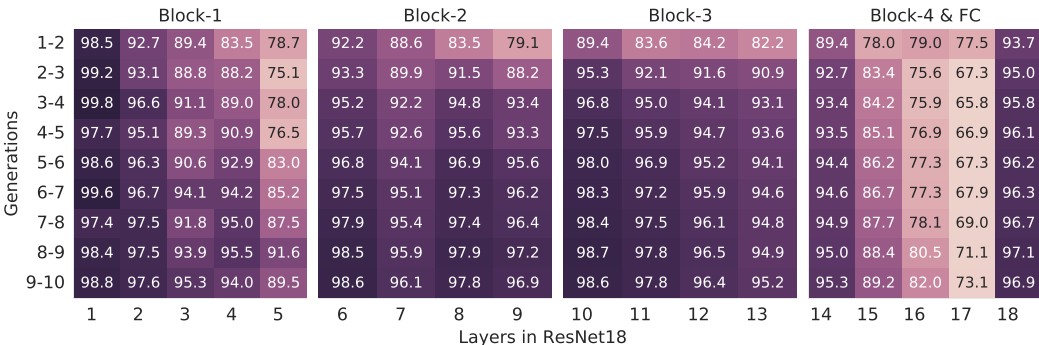

Figure 4: Layer-wise percentage overlap of the retained parameters in consecutive generations.

baseline. This highlights the efficacy of our framework in training models that can learn high-level abstractions robust to small perturbations in the input data.

## 5.3 ROBUSTNESS OF CONNECTION SELECTION ACROSS TRAINING STEPS

Unlike KE, which employs a randomly predetermined and fixed masking strategy, SKE provides a notable advantage through the utilization of Data-aware Dynamic Masking (DDM) for parameter reinitialization. Therefore, it is crucial to investigate whether SKE fully leverages the benefits of the evolutionary training paradigm by dynamically adapting the mask in each generation.

The proposed SKE framework employs SNIP (Lee et al., 2018)) as a masking criterion to selectively regulate the parameters that have the least impact on performance at each generation of training. To examine this, we analyze the CUB200 dataset using the ResNet18 architecture. We save the mask generated by SNIP after the end of every generation. Visualizing the mask generated by the SKE framework can be challenging due to the large number of parameters in each layer of the backbone. To assess the consistency of connections across generations, we adopt a metric based on the percentage of overlap of retained parameters between the masks created in consecutive generations. This metric provides a quantitative analysis of the degree of flexibility induced by SKE in the evolutionary training process.

Figure 4 illustrates the layer-wise percentage overlap of retained parameters between consecutive generations in the SKE framework. The results reveal that the earlier layers consistently exhibit a high overlap percentage across all generations, indicating a consistent selection of connections.

The overlap percentage decreases in the later layers (specifically, layer 4 in ResNet) as the model learns class-specific information. This observation suggests that the mask adapts to capture task-specific features while maintaining stability in the earlier layers. Interestingly, we observe that the overlap percentage of the mask progressively increases as the evolutionary training progresses. Specifically, the overlap between the 9th and 10th generations is higher compared to the overlap between the 1st and 2nd generations. This

Table 3: Evaluating the performance of SKE with different importance estimation.

| Importance Criteria | CUB200 | Flower |
|---|---|---|
| LB | 69.80 $\pm 0.13$ | 66.89 $\pm 0.23$ |
| Random (KE) | 66.51 $\pm 0.07$ | 62.56 $\pm 0.17$ |
| FIM | 67.73 $\pm 0.28$ | 65.96 $\pm 0.20$ |
| Weight Magnitude | 64.18 $\pm 0.19$ | 66.90 $\pm 0.11$ |
| SNIP | **71.87** $\pm 0.22$ | **68.36** $\pm 0.14$ |

observation suggests that the mask becomes more saturated and stable as the model state converges to a lower-loss landscape. This flexible nature of the SKE framework, allowing for the regulation of connections in both early and later layers, contributes to its effectiveness in improving generalization performance.

## 5.4 EFFECT OF IMPORTANCE ESTIMATION METHOD

We conducted an investigation into the effectiveness of different methods to estimate the importance of parameters within our proposed training paradigm. Specifically, we explore the Fisher Importance (FIM), weight magnitude, random selection, and SNIP (Lee et al., 2018) criteria. In Table 3, we present the performance and generalization results of the model trained with these various selection methods on the CUB200 dataset using the ResNet18 architecture.

Our findings demonstrate that the use of SNIP as data-aware dynamic masking yields superior performance compared to all other baseline methods. Surprisingly, the importance criteria based on weight magnitude exhibited inferior performance compared to random selection. However, the lottery ticket hypothesis (Frankle & Carbin, 2018) suggests the existence of sparse subnets within neural networks. Remarkably, when these subnets are trained in isolation, they can achieve a final performance accuracy comparable to that of the entire network in the same or even fewer training epochs. In particular, neurons within these winning subnets demonstrate higher rates of weight changes relative to other neurons. This observation raises the possibility of selectively reinitializing neurons that undergo minimal weight changes during training, as they contribute the least to loss function. Merely relying on the $\ell_1$ norm, which fails to capture the rate of weight changes, as described by the lottery ticket hypothesis, may not adequately capture the notion of importance. Therefore, our findings suggest that the utilization of SNIP for data-aware dynamic masking proves to be more effective, as it considers the rate of weight changes in determining the importance of parameters. This approach aligns better with the lottery ticket hypothesis and leads to improved performance and enhanced generalization capabilities in our experimental evaluations.

## 5.5 EFFECT OF VARYING THE RATIO OF REINITIALIZED PARAMETERS.

Table 4 shows the effect of varying the number of reinitialized parameters on the performance and generalization of the model. We train the model in evolutionary settings using the SKE framework by varying different percentages of reinitialized parameters (5%, 10%, 20%, 30%, and 40%). Experiments were carried out with ResNet18. The results show that the reinitialization of a 5% percentage of parameters has no impact on performance, while reinitialization of more than 30% has less impact on test accuracy. We find that reinitialization 20% of the parameters results in the best performance.

Table 4: Performance evaluation with varying the percentage of reinitialized parameters during training using ResNet18. Test accuracy at the end of 10 generations is reported on Aircraft and CUB datasets.

| Reinit. Params (%) | Aircraft | CUB |
|---|---|---|
| 5 | 65.34 | 69.95 |
| 10 | 66.10 | 70.15 |
| 20 | **66.63** | **71.37** |
| 30 | 64.13 | 68.42 |
| 40 | 62.79 | 66.87 |

## 6 CONCLUSION

We present Selective Knowledge Evolution (SKE), an iterative/evolutionary training paradigm designed to improve the generalization of deep networks on small datasets. Our framework incorporates selective neurogenesis at the end of each generation, employing a data-aware dynamic masking scheme to remove redundant connections according to their importance. This enables the model to increase its capacity for further learning through random weight reinitialization, emphasizing the acquisition of generalizable features. Empirical results demonstrate that the proposed framework substantially enhances performance and generalization across small datasets, achieving comparable results on large-scale datasets compared to other reinitializing techniques. Moreover, SKE exhibits improved robustness in challenging real-world scenarios, including adversarial attacks and learning with class imbalances, while enhancing generalization on natural corruption data. Additionally, exploring the potential of growing networks presents an intriguing avenue for future research. Finally, our intention was to first demonstrate the practical effectiveness of our proposed method. We hope that theoretical advancements on this topic will be subjects of future study.

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

# A APPENDIX

## A.1 EVOLUTION OF MASK ACROSS GENERATIONS

In this section, we present the evolutionary process of the mask over multiple generations and its impact on the performance and generalization capabilities of the DNNs. We evaluate the effectiveness of our proposed method, Selective Knowledge Evolution (SKE), in dynamically adapting and evolving the mask throughout the training process. The ResNet18 architecture with CUB200 is used for this evaluation.

Figure 5 illustrates the evolution of the mask across generations. As the training progresses, the mask undergoes iterative updates based on the data-aware dynamic masking criteria employed by SKE. The mask becomes more refined and selective with each generation, preserving important connections while pruning less relevant ones.

Figure 5: Layer-wise percentage overlap of the retained parameters between first and corresponding generations.

To quantify the evolution of the mask, we measure the overlap percentage of parameters retained between the first and the corresponding generations. We observe a gradual decrease in overlap from the initial generation to subsequent generations, indicating the emergence of masks in an evolutionary training scenario. This progressive mask evolution contributes to the network's enhanced capacity for learning and generalization, evident from the test accuracy.

In conclusion, our results highlight the evolutionary nature of the mask throughout generations in the SKE framework. The dynamic adaptation and refinement of the mask lead to effective masking and improved performance and generalization of the DNN. These findings support the effectiveness of our approach in leveraging the evolutionary training paradigm to enhance the learning and generalization capabilities of deep neural networks compared to KE.

---

**Algorithm 1** Selective Knowledge Evolution (SKE)

> **input:** Train Data $D_t \ \forall \ t \ \in \{1, ..., T\}$, Model $f_\Theta$.
> Sparsity factor $k$, learning rate $\eta$, Binary Mask $M$, parameters $\Theta$, Small subset of dataset $(\pi = 0.2|D_t|)$

1: **for all** Generation $g \in \{1, 2, .., N\}$ **do**
2:     $f_g \leftarrow$ Train $f_\Theta$ for e epochs with learning rate $\eta$;              ▷ Training step
3:     $M \leftarrow$ Importance Estimation$(f_g, \pi, k)$
4:     Retain the task specific weights based on $M$           ▷ Knowledge Selection
5:     Randomly reinitialize the non-important parameters in $f_g$.
6:     Model with this new initialization for next generation training

---

## A.2 DIFFERENCE WITH TRANSFER LEARNING

Our approach, Selective Knowledge Evolution (SKE), indeed differs widely from the domain of transfer learning. Unlike transfer learning, which primarily focuses on leveraging pre-trained mod-

Table 5: Comparison of SKE with transfer learning.

| Baselines | CUB | Aircraft | Dog | Flower |
|---|---|---|---|---|
| Smth + transfer learning (f3) | 65.63 ±0.21 | 61.02 ±0.23 | 63.84 ±0.17 | 57.62 ±0.19 |
| Smth + SKE (f3) | **68.56** ±0.24 | **64.37** ±0.19 | **65.72** ±0.15 | **62.13** ±0.23 |

els trained on large datasets from different domains to boost task performance on downstream tasks, SKE is intricately designed to tackle the intricate challenge of enhancing generalization in the presence of inherently limited or small datasets. A key issue with transfer learning arises when the pre-trained model's source domain vastly differs from the target domain of interest. This discrepancy between domains often leads to domain shifts, where the knowledge transferred from the pre-trained model fails to adapt well to the specificities of the target domain, thereby resulting in suboptimal performance.

In particular, in scenarios like medical applications, obtaining sufficient labeled data that closely aligns with the task at hand is exceptionally challenging. Though transfer leaning is predominantly used in this field, the need for domain expertise, privacy concerns, and the uniqueness of each application domain make it exceedingly difficult to find a pre-trained model that seamlessly fits. Furthermore, the presence of domain shift between the source and target might lead to compromised performance affecting the accuracy and generalization of the model on a specific task with limited data.

SKE, on the other hand, offers a novel solution to these intricate challenges. By employing data-aware dynamic masking and selective reinitialization, SKE fosters the gradual evolution of the network, enabling it to adapt more effectively to the characteristics of the specific dataset. This process circumvents the problems of domain shifts that often plague transfer learning methods. Thus, while transfer learning remains valuable in contexts with abundant and well-aligned data, SKE stands out as a specialized approach to address the unique hurdles faced in scenarios of limited data availability, where the domain shift problem can severely hinder model performance and generalization.

Furthermore, we have included a comparative analysis in Table 5 involving an instance of transfer learning within the iterative training process. In this particular case, weights are directly transferred from one generation to the next without undergoing reinitialization.

This comparison serves to highlight the unique effectiveness of the Selective Knowledge Evolution (SKE) method. Our results distinctly demonstrate that SKE enhances the process of generalization, showcasing superior performance in comparison to the approach of directly transferring the complete network's weights across generations. This outcome further underscores the distinct advantage of SKE in evolving the network's capacity for better adaptation and learning in the evolutionary training paradigm.

### A.3 CONVERGENCE BEHAVIOR OF SKE COMPARED TO TRANSFER LEARNING

In Figure 6, we present the convergence behaviour of the Selective Knowledge Evolution (SKE) algorithm juxtaposed with vanilla fine-tuning. The x-axis delineates different generations during the training process, while the y-axis represents the performance at the end of each generation. We observe that the convergence of vanilla fine-tuning unfolds at a more gradual pace. In contrast, SKE demonstrates a faster convergence rate. Across generations, SKE consistently surpasses vanilla fine-tuning, delivering a heightened performance level within a shorter training duration. The utilization of data-aware dynamic masking through

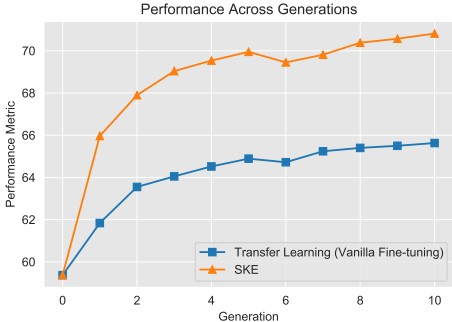

Figure 6: Convergence Behavior of SKE with ResNet18 trained on CUB dataset

Table 6: Additional comparison with the layerwise reinitialization methods.

| Baselines | CUB | Aircraft | Dog | Flower |
|---|---|---|---|---|
| LW (N8) | 70.50 $\pm 0.26$ | 67.10 $\pm 0.32$ | 65.76 $\pm 0.36$ | 66.92 $\pm 0.20$ |
| LLF (N8) | **71.30** $\pm 0.14$ | **68.87** $\pm 0.12$ | 66.35 $\pm 0.22$ | 67.20 $\pm 0.24$ |
| SKE (N8) | 70.87 $\pm 0.16$ | 66.10 $\pm 0.25$ | **66.56** $\pm 0.18$ | **68.50** $\pm 0.27$ |

SNIP in SKE amplifies this efficiency, enabling the model to concentrate on the most pertinent information for effective generalization.

## A.4 ADDITIONAL COMPARISON WITH THE LAYERWISE REINITIALIZATION METHODS

For a more thorough evaluation, we compare SKE with the layerwise reinitialization methods and provide a detailed comparison to showcase the advantages and uniqueness of our proposed approach.

Zhou et al. (2022) (LLF) propose the forget and relearn hypothesis, which aims to harmonize various existing iterative algorithms by framing them through the lens of forgetting. This approach operates on the premise that initial layers capture generalized features, while subsequent layers tend to memorize specific details. Accordingly, they advocate for the repeated reinitialization and retraining of later layers, effectively erasing information related to challenging instances. Similarly, the LW (Alabdulmohsin et al., 2021) approach progressively reinitializes all layers. Table 6 demonstrates a comparison with these methods.

Notably, SKE showcases comparable, or slightly enhanced performance compared to LW and LLF. Also, these methods (LLF, LW) are underpinned by architecture-specific assumptions that are independent of the data. They rely on the assumed properties that are inherent to the model and its learning. These methods lack a priori knowledge of where and what features, layers, etc. should be reinitialized in general settings. Furthermore, as the model's architecture scales, the complexity of these methods increases accordingly, potentially leading to scalability challenges like which layers to reinitialize. Our proposed method, in contrast, leverages data-aware connection sensitivity through the employment of SNIP, enabling us to select connections for reinitialization dynamically based on their redundancy, contributing to improved generalization.

## A.5 ROBUSTNESS TO CLASS IMBALANCE DATASET

In real-world applications, class imbalance is a common characteristic of the input distribution, where certain classes are more prevalent than others. This inherent class imbalance can affect the training of DNNs, as they tend to be biased towards the majority classes, thereby neglecting the minority classes (Chrysakis & Moens, 2020). To address this issue, we explore the contribution of reinitialization to model training with class imbalance. We incorporate class imbalance using the power law model on CIFAR10. The number of training samples for a class $c$ is determined by the formula $n_c = a/(b + c)^\gamma$, where $\gamma$ represents the imbalance ratio, and $a$ and $b$ are offset parameters specifying the largest and smallest class sizes. We set a fixed gamma value of 1 in our experiments to maintain a power law class distribution. The offset parameters $a, b$ are chosen such that the maximum and minimum class counts are 5000 and 250, respectively. We used balaced accuracy as a metric to measure the robustness of the model under class imbalance scenario. Our findings in Figure 7 demonstrate that the SKE

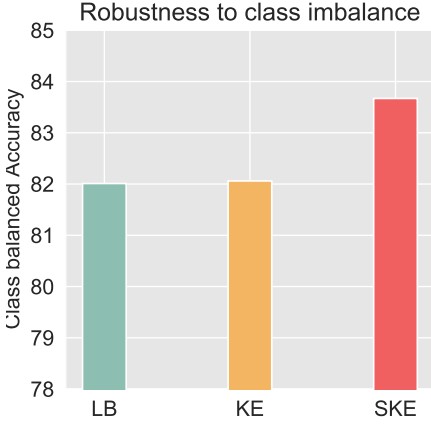

Figure 7: Robustness to Class Imbalance

framework consistently outperforms KE and longer baselines in scenarios with class imbalance. This highlights the effectiveness of SKE in addressing the challenges posed by imbalanced class distributions and underscores its potential for practical applications.

## A.6    EVALUATING THE EFFECTIVENESS OF THE SPARSE MODEL

In this section, we assess the effectiveness of the sparse model (containing 20% fewer parameters than the full/dense model) obtained through the selective neurogenesis process during the inference phase. We examine the sparse and dense models' test performance compared to the original KE framework. For this, we measure the performance of the ResNet18 model trained on CUB200. Table 7 presents the accuracy results obtained by the sparse model compared to the dense model. Surprisingly, despite the considerable reduction in the number of parameters, the sparse model achieves comparable accuracy compared to the dense model in the SKE framework. Furthermore, SKE demonstrates superior performance in both the full and sparse model scenarios compared to the KE. This indicates that the selective neurogenesis process successfully retains the critical connections necessary for accurate predictions while eliminating redundant or less informative connections. Our evaluation demonstrates that the sparse model obtained through the selective neurogenesis process offers several benefits during inference. It maintains high accuracy while achieving improved computational efficiency compared to the KE. These results highlight the practicality and efficacy of leveraging selective neurogenesis for creating efficient and compact deep learning models that can be readily deployed in real-world scenarios.

Table 7: Evaluating the effectiveness of the sparse model.

| Method | Full model | Sparse model |
|---|---|---|
| KE ($f_{10}$) | 66.51 | 66.21 |
| SKE ($f_{10}$) | **71.37** | **70.08** |

## A.7    VARYING THE QUANTITY OF DATA USED FOR IMPORTANCE ESTIMATION

In our experiments, we randomly sampled 20% of the dataset to estimate the importance of the parameters after the end of each generation. Here, we analyze the impact of the number of data used to determine the important estimation on the final performance. Similar to Lee et al. (2018), we used as few as 128 samples to estimate the important parameters using SNIP. Table 8 shows that SKE is not sensitive to the variation in the input data used to estimate the importance as the final performance remains unchanged.

## A.8    SUMMARY OF DATASETS AND IMPLEMENTATION DETAILS

Taha et al. (2021) employs various image resizing techniques for different datasets; however, they do not provide specific details about the resizing parameters in their paper. To ensure consistency across our experiments, we resize all datasets to a fixed size of (256, 256). Moreover, to fine-tune the hyperparameters, we utilize a validation split, and the reported results are based on the test set whenever it is available.

For experiments on large datasets, we used the following settings. The experiments were conducted on three different datasets: CIFAR-10/100, Tiny-ImageNet. For CIFAR-10/100, the training was performed for 160 epochs. A batch size of 64 was used, along with a step-based learning rate scheduler. The learning rate decay was applied between epochs 80 and 120, with a decay factor of 10. The momentum was set to 0.9, and l2 regularization was applied with a coefficient of 5e-4. Initial learning rate used was 0.1. There were no warmup epochs in this case.

For the Tiny-ImageNet dataset, the training was also conducted for 160 epochs. The batch size was reduced to 32, and a step-based learning rate scheduler was used. Similar to CIFAR-10/100, the learning rate decay occurred between epochs 80 and 120, with a decay factor of 10. The momentum and l2 regularization were set to 0.9 and 5e-4, respectively. Additionally, 20 warmup epochs were applied. Throughout all experiments, a resetting ratio of 20% is used for all generation. All the training and evaluation is done on NVIDIA RTX-2080 Ti GPU. The time required to approximately

Table 8: Evaluation with varying the quantity of data for importance estimation. Test accuracy at the end of 10 generations is shown on Aircraft and CUB datasets.

|  | # samples | Aircraft | CUB |
|---|---|---|---|
| SKE | $0.2\,|\mathcal{D}|$ | **66.63** | **71.37** |
|  | 128 | 66.45 | 71.26 |

train 10 generation of SKE on CUB200 with ResNet18 is approximately 1.68 hours. It's worth mentioning that for comparing our method with other baselines, we utilized the results presented in the KE paper (Taha et al., 2021) as a point of reference. For the hyperparameters used in training small datasets, please refer to Section 4.

Table 9: shows the statistics of five classification datasets.

| Datasets | Classes | Train | Validation | Test | Total |
|---|---|---|---|---|---|
| CUB-200 (Wah et al., 2011) | 200 | 5994 | N/A | 5794 | 11788 |
| Flower-102 (Nilsback & Zisserman, 2008) | 102 | 1020 | 1020 | 6149 | 8189 |
| MIT67 (Quattoni & Torralba, 2009) | 67 | 5360 | N/A | 1340 | 6700 |
| Aircraft (Maji et al., 2013) | 100 | 3334 | 3333 | 3333 | 10000 |
| Standford-Dogs (Khosla et al., 2011) | 120 | 12000 | N/A | 8580 | 20580 |

