# OpenReview forum: "Improving Generalization for Small Datasets with Data-Aware Dynamic Reinitialization"
_ICLR.cc/2024/Conference — Submitted to ICLR 2024_

### Official Review · Reviewer_U7Q7 · 2023-10-26

**Soundness:** 2 fair
**Presentation:** 3 good
**Contribution:** 2 fair
**Rating:** 6
**Confidence:** 3

**Summary:**

This paper studied how to train a neural network on small datasets while improving generalization. Motivated by neurogenesis in the brain, this paper proposed a novel iterative training framework, Selective Knowledge Evolution (SKE), that employs a stage-wise mask to reinitialize the masked weights per training stage. The stage-wise mask is obtained by estimating the data-dependent sensitivity via SNIP. The proposed SKE shows impressive empirical improvements in various experiments.

**Strengths:**

-	This work made two simple but effective modifications to the original Knowledge Evolution (KE) method. Both the neuroscience-inspired dynamic mask and the selective mask via SNIP are interesting.
-	I appreciate the part which introduces the inspiration from neurogenesis in the brain.
-	The empirical improvements seem significant and general in various experiments.
-	The experiments are comprehensive and beyond simple accuracy comparison on small datasets.

**Weaknesses:**

-	The work did not study the computational cost comparable with KE. Moreover, I believe the empirical improvement will more convinceable if the authors may also compare the generalization under similar computational costs. Because it is known that DNNs sometimes improve generalization with longer training.
-	There is theoretical analysis at all. Theoretical understanding under some assumptions will be appreciated.

**Questions:**

Please see the weaknesses.

---

> ### Author Response · Authors · 2023-11-22
> **Reply to Reviewer U7Q7**
>
> We would like to thank the reviewer for their valuable feedback and acknowledgment of the simplicity and effectiveness portrayed in our work. We are particularly grateful for the highlighted strengths.
>
> > The work did not study the computational cost comparable with KE. Moreover, I believe the empirical improvement will more convinceable if the authors may also compare the generalization under similar computational costs. Because it is known that DNNs sometimes improve generalization with longer training.
>
> In our experiments, we consistently maintained a fixed training duration of 200 epochs for each generation, with the number of generations set at 10 for fair comparison. The computational cost of evolutionary training methods, including both KE and SKE, scales linearly with the number of generations (T). For example, if KE is trained for 5 generations, the total computational cost becomes 5T times that of training a single generation. Similarly to ensure fairer comparison, we train a long baseline for the same number of epochs.
> The additional computational cost incurred by SKE for computing data-aware dynamic masking with SNIP is minimal. For instance, on the CUB dataset with a 20% subset, it amounts to 20.3 seconds per generation. This is a one-time calculation performed at the end of each generation. This computational cost can be further reduced by using just 128 samples to estimate the importance without affecting the final performance. Notably, SKE's performance exhibits minimal sensitivity to changes in the subset size, as demonstrated in Appendix, Table 8.
>
> Overall, SKE maintains an equivalent computational cost compared to our long baselines and KE. The slight increase in computational cost in SKE, attributed to weight reinitialization and computing connection sensitivity after each generation, is justified by the substantial improvement in generalization performance. We believe this detailed clarification adequately addresses concerns about computational costs satisfactorily.
>
> > There is theoretical analysis at all. Theoretical understanding under some assumptions will be appreciated.
>
> This paper is indeed an empirical paper. While theoretical analysis is essential, our intention was to first demonstrate the practical effectiveness of our proposed method. We hope that theoretical advancements on this topic will be subjects of future study. In general, theoretical studies on why reinitialization during standard training improves generalization are not well explored. We will make a note of this in our future work section, highlighting the need for deeper theoretical analysis to complement our empirical findings.

---

> > ### Comment · Reviewer_U7Q7 · 2023-11-23
> > **Acknowledge the responses**
> >
> > Thanks for the responses.
> >
> > I plan to keep the original rating.

---

> > > ### Author Response · Authors · 2023-11-23
> > > **Response to Reviewer U7Q7**
> > >
> > > Thank you for taking the time to revisit our discussion and for considering the points we addressed in response to your concerns. We appreciate your dedication to this matter.
> > >
> > > We've diligently worked to address the issues and incorporate your valuable feedback into our processes. However, we find ourselves perplexed as to the specific reason behind your decision to uphold the original rating. We hope you can shed further light on this decision, as it will assist us in comprehending and addressing any remaining concerns.
> > >
> > > Thank you once again for your time and collaboration.

---

### Official Review · Reviewer_p2E7 · 2023-10-30

**Soundness:** 2 fair
**Presentation:** 2 fair
**Contribution:** 2 fair
**Rating:** 5
**Confidence:** 4

**Summary:**

This paper builds upon previous studies that examined the effects of reinitialization on the generalization capabilities of neural networks. The authors focus specifically on ResNets and small datasets (e.g. of size < 100K examples). Earlier research highlighted that by selectively reinitializing certain network parameters and then retraining the whole network, one could enhance generalization. The current paper presents a new approach for choosing which parameters to reset that is data-dependent, and show that it outperforms several baseline methods. Additionally, the authors provide additional useful analysis, such as the consistency in parameter selection across iterations, comparisons with other importance estimation techniques, and robustness-related evaluations.

**Strengths:**

- The authors explore an interesting direction: to reinitialize the network during training to improve its generalization capability. This has been studied in several recent works, and an additional investigation can be useful to the community.
- The paper shows notable improvements in performance compared to *some* baselines (namely, KE, DSD, long training). As mentioned in the paper, this approach has the potential of improving generalization in data-scarce domains, such as in healthcare.

**Weaknesses:**

- It is not clear how statistically reliable the main conclusions of the paper are. This is because the number of datasets used is small (e.g. the authors evaluate ResNet18 on only four small datasets). Since the datasets are small and there are tens of vision-related datasets available, I'm curious to know why the authors used only four datasets. I don't think including more datasets should be an issue since (again) they are small and ResNet18 is also small. The same applies to the other sections. For example, the authors evaluate robustness to adversarial attacks in one dataset only!

- The authors only compare with old reinitialization baselines (KE, DSD, and BAN). There have been more recent layerwise methods, but the authors only compare against those in the appendix (see Table 5). There, the improvement is marginal. Is there a reason the authors chose to not include those other methods in the main paper? Also, why aren't they included in the ResNet50 evaluation? They are also not discussed in the Related Works section, and there are a few recent related works missing as well; e.g. (Sheheryar, et al. 2023) and (Jaehoon, et al. 2022). They should all be discussed in the related works section.

- The reported results for CIFAR10-C seem too low to me. I would expect ResNet18 trained on CIFAR10 to have an accuracy larger than 60% when evaluated on CIFAR10-C.

- The authors argue in the appendix that transfer learning is not useful for medical applications. But that's not true. Many SotA models are pretrained on datasets, like ImageNet. See for example: https://www.nature.com/articles/s41591-020-0842-3.

- There are a few places containing typos, incomplete sentences, or undefined symbols:
   * Page 2: "Finally, Our work ... " --> "Finally, our work ..."
   * Equation 4: $\pi$ is undefined.
   * Page 4: "Due to the difficulty in deciding ..." is not a complete sentence.
   * Page 4: "retained learned" should be either "retained" or "learned".

**Questions:**

- When the authors compare against long-baseline (LB), do they also remove 20% of the examples in LB? I'm asking this because LB does not need to have 20% of the examples removed, unlike SKE which uses those for importance estimation.
- Please explain precisely how the Mean Corruption Accuracy metric is calculated?

---

> ### Author Response · Authors · 2023-11-22
> **Reply to Reviewer p2E7 (1/2)**
>
> > It is not clear how statistically reliable the main conclusions of the paper are. This is because the number of datasets used is small (e.g. the authors evaluate ResNet18 on only four small datasets). Since the datasets are small and there are tens of vision-related datasets available, I'm curious to know why the authors used only four datasets. I don't think including more datasets should be an issue since (again) they are small and ResNet18 is also small.
>
>
> Thank you for highlighting the concern about the number of datasets in our study. In response, we have expanded the empirical validation of our method by including results on a new dataset. The revised paper now encompasses evaluations on a total of five small datasets and three large datasets—CIFAR-10, CIFAR-100, and Tiny ImageNet.
>
> This allows us to present a more comprehensive view of the performance of Selective Knowledge Evolution (SKE) across a diverse set of datasets. Notably, in the majority of these datasets, SKE consistently outperforms both Knowledge Evolution (KE) and the longer baseline. These results affirm that SKE brings discernible benefits in terms of improving generalization.
>
> It's worth mentioning that the choice of datasets aligns with the baselines established in the original paper, ensuring a fair comparison and leveraging the availability of their results. Also, we are incorporating results on adversarial perturbation for a new dataset in the revised version of our paper. This addition aims to provide a more comprehensive understanding of the performance and robustness of Selective Knowledge Evolution (SKE) across datasets under different scenarios.
>
> If you have any further questions or if there are specific aspects you would like more clarification on, please let us know.
>
> > The authors only compare with old reinitialization baselines (KE, DSD, and BAN). There have been more recent layerwise methods, but the authors only compare against those in the appendix (see Table 5). There, the improvement is marginal. Is there a reason the authors chose to not include those other methods in the main paper?
>
> The rationale behind featuring comparisons with more recent layerwise methods in Appendix, rather than the main body of the paper, stems from our strategic emphasis on elucidating the distinctive merits and innovations of Selective Knowledge Evolution (SKE). SKE stands out by addressing the entire network, in contrast to layerwise methods like LW and LLF that operate on specific layers with architecture-specific assumptions. Thus, in the main paper, our priority was to underscore the advantages over reinitialization baselines like KE, DSD, and BAN.
> In Appendix, we thoughtfully presented additional comparisons, encompassing recent layerwise methods. The marginal improvement observed in these comparisons reinforces the effectiveness of SKE, demonstrating comparable or slightly enhanced performance. Importantly, this decision does not diminish the significance of these comparisons but aligns with our intent to maintain focus in the main paper. The inclusion in the appendix ensures that interested readers can delve into supplementary insights without detracting from the core contributions highlighted in the main text.
>
> >  They are also not discussed in the Related Works section, and there are a few recent related works missing as well; e.g. (Sheheryar, et al. 2023) and (Jaehoon, et al. 2022). They should all be discussed in the related works section.
>
> We acknowledge the importance of addressing recent related works suggested by the reviewer, including (Sheheryar, et al. 2023) and (Jaehoon, et al. 2022). In the revised manuscript, we incorporated discussions of these works in the Related Works section to ensure a comprehensive overview of the literature in the field.
>
> > The authors argue in the appendix that transfer learning is not useful for medical applications. But that's not true. Many SotA models are pretrained on datasets, like ImageNet. See for example: https://www.nature.com/articles/s41591-020-0842-3.
>
> We want to clarify that our intention was not to claim that transfer learning is not useful for medical applications. Instead, we aimed to highlight a limitation often associated with transfer learning – domain shift. In situations with limited datasets, domain shifts can significantly impact performance. We acknowledge the importance of transfer learning in various contexts, including medical applications, and we adjusted the corresponding statements in the revised paper to better convey this nuanced perspective.
>
> > There are a few places containing typos, incomplete sentences, or undefined symbols
>
> We want to acknowledge that these issues have been duly rectified in the revised version of the manuscript. Thank you for your meticulous review.

---

> ### Author Response · Authors · 2023-11-22
> **Reply to Reviewer p2E7 (2/2)**
>
> > When the authors compare against long-baseline (LB), do they also remove 20% of the examples in LB? I'm asking this because LB does not need to have 20% of the examples removed, unlike SKE which uses those for importance estimation.
>
> We do not exclude the subset of data (pi) used to calculate the importance estimate; it is retained throughout. We solely utilize 20 percent of the data for estimating the importance for SKE. Training is consistently conducted with the entire dataset, encompassing 100 percent of the data, for all our experiments, including SKE.
>
> > The reported results for CIFAR10-C seem too low to me. I would expect ResNet18 trained on CIFAR10 to have an accuracy larger than 60% when evaluated on CIFAR10-C. Please explain precisely how the Mean Corruption Accuracy metric is calculated?
>
> Thank you for your insightful observation. The reported results for CIFAR10-C are indeed based on a model trained on the CUB dataset and evaluated on CIFAR10-C as part of our worst-case out-of-domain robustness analysis, rather than the standard CIFAR10 dataset. This distinction has now been made explicit in the revised version of our paper to avoid any potential confusion. Additionally, we will include the results of a model trained on CIFAR10 and evaluated on CIFAR10-C in the revised version.
>
> The Mean Corruption Accuracy (mCA) metric is calculated as follows:
>
> $mCA = \frac{1}{{N_c \times N_s}} \sum_{c=1}^{N_c} \sum_{s=1}^{N_s} A_{c, s}$
>
> Where $N_c$ represents the number of corruptions (in this case, 19), and $N_s$ represents the number of severity levels (in this case, 5). The variable $​A_{c, s}$  denotes the F1-score measure evaluated on CIFAR-C under the c-th corruption with the s-th severity level. The mCA metric provides an average performance measure across all corruptions and severity levels, offering a comprehensive evaluation of the model's robustness under different forms and degrees of corruption.

---

### Official Review · Reviewer_a3Fd · 2023-10-31

**Soundness:** 3 good
**Presentation:** 3 good
**Contribution:** 3 good
**Rating:** 6
**Confidence:** 4

**Summary:**

The authors present a new method to improve training of neural networks. The method involves reinitializing a subset of the weights, selected based on saliency, a number of times during training. The authors compare their method to similar knowledge evolution experiments through evaluation on performance, corruptions, adversarial attacks and imbalanced datasets.

**Strengths:**

- The method presented is an interesting research direction that ties together many recent research directions, e.g. pruning, reinitialization and evolution. This gives it a reasonable motivation.
- The paper itself is well-structured and mostly well written (but see exceptions listed below)
- Experiments suggest the method is promising although more investigation are needed. They will be of interest to the research community.

**Weaknesses:**

- The main issue with the paper is its presentation of the results on imbalanced data. The authors use accuracy as a metric, which can be highly misleading on imbalanced data. I cannot infer anything from figure 4 on the robustness to imbalanced data. There are many ways to show performance on imbalanced data, including confusion matrices or per-class accuracy. I suggest the authors either improve this presentation or remove entirely the class imbalance experiments (or correct the figure and text description if they do not mean overall accuracy). In its current form the results are not supporting the conclusions.
- The adversarial perturbation experiments are also lacking in presentation. Is the figure a single run or multiple runs? What are the errors?
- Sensitivity analysis would have been a nice addition, the authors leave this for future work. However, the paper would have been notably stronger with these included and should be relatively inexpensive to run.

**Questions:**

Questions:
- In the abstract and introduction, the authors state that the KE approach is limited due to its predetermined mask. I can see that this as a valid hypothesis, but the authors state this as if it is well-established. Are there any references to support this? With the posterior knowledge of the results of SKE this statement is supported, but the hypothesis which sparked the investigation cannot be based on the results.
- How is the size of subset used to evaluate connection sensitivity selected? Is there any estimate for what is sufficient?
- Sensitivity analysis is not part of this paper but how do the authors interpret the sparsity constraint k? Is there anything in the literature or insight they have as to how sensitive the model is to it?

Other comments:
- There are instances where the word "significant" is used to describe the difference between two methods. I highly recommend that the authors save this term to describe statistical significance (it seems a significance test was not performed). There are better words to describe a great difference that are not as ambiguous.
- It should be made much clearer that section 3.1 is standard KE and not the version being presented in the paper. On first reading it seems like it is the method being introduced and the fixed mask creates confusion.
- When the binary mask is defined in 3.1, make it clear it is binary when it is first mentioned, not in a later paragraph.
- Grammatical error: "We define a deep neural network f with L layers and is characterized by"
- The acronym DNN is defined multiple times. The authors should define it once at the start and then only use it and not spell it out or redefine it multiple times.
- Figure 4 is never referenced in the main text

**Details Of Ethics Concerns:**

There are no obvious ethical concerns. It might be possible to say that this method could introduce further bias by being data-aware. Bias in the data could affect the weight samples and therefore the model. However, it is not obvious if this would be significantly different to biases introduced through backpropagation, which is data-aware itself.

It would be better if the authors acknowledge this (and any other forms of potential bias they identify), but in the context of this method I do not see it as critical.

---

> ### Author Response · Authors · 2023-11-22
> **Reply to Reviewer a3Fd (1/2)**
>
> >  The main issue with the paper is its presentation of the results on imbalanced data. The authors use accuracy as a metric, which can be highly misleading on imbalanced data. I cannot infer anything from figure 4 on the robustness to imbalanced data. There are many ways to show performance on imbalanced data, including confusion matrices or per-class accuracy. I suggest the authors either improve this presentation or remove entirely the class imbalance experiments (or correct the figure and text description if they do not mean overall accuracy). In its current form the results are not supporting the conclusions.
>
> We appreciate your insightful comment. We would like to clarify that the accuracy mentioned in the context of class imbalance is indeed the balanced accuracy. First we evaluate the Recall for each class, then we average the values in order to obtain the Balanced Accuracy score. We have taken steps to enhance the clarity of our presentation in the revised version. Moreover, we have reorganized of our paper based on your suggestions. The robustness to class imbalance experiments has been moved to the appendix (Appendix Section A5), allowing us to highlight the significance of sensitivity analysis in the main paper (Section 5.5). We trust that this adjustment will contribute to a clearer and more effective presentation of our findings.
>
>
> > The adversarial perturbation experiments are also lacking in presentation. Is the figure a single run or multiple runs? What are the errors?
>
> To address this concern, we modified the figure to include error bars to provide a more comprehensive representation of the experimental results.
>
>
> > Sensitivity analysis would have been a nice addition, the authors leave this for future work. However, the paper would have been notably stronger with these included and should be relatively inexpensive to run.
>
> Thank you for your valuable suggestion regarding the inclusion of sensitivity analysis in our paper. Table 4 shows the effect of varying the number of reinitialized parameters on the performance and generalization of the model. We train the model in evolutionary settings using the SKE framework by varying different percentages of reinitialized parameters (5%, 10%, 20%, 30%, and 40%). Experiments were carried out with ResNet18. The results show that the reinitialization of a 5% percentage of parameters has no impact on performance, while reinitialization of more than 30% has less impact on test accuracy. We find that reinitialization 20% of the parameters results in the best performance.  We agree that this would be a valuable addition to strengthen our work, and we appreciate your insight.
>
> > In the abstract and introduction, the authors state that the KE approach is limited due to its predetermined mask. I can see that this as a valid hypothesis, but the authors state this as if it is well-established. Are there any references to support this? With the posterior knowledge of the results of SKE this statement is supported, but the hypothesis which sparked the investigation cannot be based on the results.
>
> Thank you for raising this point. The motivation drawn from neurogenesis indeed inspired our exploration into the impact of a fixed mask in the context of deep neural networks. The intricate process of neurogenesis, as highlighted in the literature (Shors et al., 2001; Garthe et al., 2016; Kempermann et al., 2015), plays a crucial role in learning and memory consolidation, allowing the brain to adapt to new experiences and stimuli.
> The specific insight into non-random integration and synaptic refinement observed in rodents' hippocampus, where the integration of new neurons leads to the elimination of less active synaptic connections (Aimone et al., 2014; Vadodaria & Gage, 2014), further fueled our curiosity. This led us to consider whether a fixed mask, akin to a predetermined synaptic connection pattern, could be limiting the adaptability and generalization capabilities of deep neural networks.
> Our exploration, guided by the emulation of selective neurogenesis, questioned the assumed fixed mask in traditional approaches such as KE. We hypothesized that a more dynamic and selective approach, akin to the characteristics of selective neurogenesis, could potentially unlock greater potential in terms of generalization in deep neural networks. Therefore, our study sought to investigate and propose a solution, Selective Knowledge Evolution (SKE), which embraces a more adaptive and dynamic approach to reinitialization, inspired by the principles of neurogenesis.

---

> ### Author Response · Authors · 2023-11-22
> **Reply to Reviewer a3Fd (2/2)**
>
> > How is the size of subset used to evaluate connection sensitivity selected? Is there any estimate for what is sufficient?
>
> To study this, we vary the quantity of data used for Importance estimation to analyze its effect on performance. In our experiments, we randomly sampled 20% of the dataset to estimate the importance of the parameters after the end of each generation. Here, we analyze the impact of the number of data used to determine the important estimation on the final performance. Similarly, we used as few as 128 samples to estimate the important parameters using SNIP. Table 8 in Appendix shows that SKE is not sensitive to the variation in the input data used to estimate the importance, as the final performance remains unchanged.
>
>
> > Sensitivity analysis is not part of this paper but how do the authors interpret the sparsity constraint k? Is there anything in the literature or insight they have as to how sensitive the model is to it?
>
> In response to your inquiry, we have incorporated additional experiments to systematically determine an appropriate subset size for evaluating connection sensitivity and the sparsity constraint k.  The results in Table 4 show that the reinitialization of a 5% percentage of parameters has no impact on performance, while reinitialization of more than 30% has less impact on test accuracy. We find that reinitialization 20% of the parameters results in the best performance.
> In the revised manuscript, we included a detailed discussion of the experimental setup for subset size selection and sensitivity in the Sections 5.5 and A7, along with the findings from these experiments.
>
>
> > Other comments:
> There are instances where the word "significant" is used to describe the difference between two methods. I highly recommend that the authors save this term to describe statistical significance (it seems a significance test was not performed). There are better words to describe a great difference that are not as ambiguous.
> It should be made much clearer that section 3.1 is standard KE and not the version being presented in the paper. On first reading it seems like it is the method being introduced and the fixed mask creates confusion.
> When the binary mask is defined in 3.1, make it clear it is binary when it is first mentioned, not in a later paragraph.
> Grammatical error: "We define a deep neural network f with L layers and is characterized by"
> The acronym DNN is defined multiple times. The authors should define it once at the start and then only use it and not spell it out or redefine it multiple times.
> Figure 4 is never referenced in the main text
>
> We extend our sincere gratitude for reviewer’s meticulous review of our manuscript. The insights are invaluable, and we implemented the suggested improvements to ensure a precise presentation of our work.

---

> > ### Comment · Reviewer_a3Fd · 2023-11-23
> >
> > I thank the authors for responding to my comments. It is good to see that some of my suggestions for improvement were implemented, however some were not (e.g. using the word significantly without doing a statistical significance test). I will be keeping my score.

---

> > > ### Author Response · Authors · 2023-11-23
> > > **Response to Reviewer a3Fd**
> > >
> > > Thank you for investing your time in reviewing the work and providing feedback. We genuinely appreciate the acknowledgment of the implemented improvements aligned with your suggestions. We have diligently addressed key concerns such as **reporting balanced accuracy** for class-imbalanced experiments, **conducting sensitivity analysis**, and **incorporating error bars to signify result averaging across multiple runs** rather than a single iteration.
> > >
> > > Regarding the use of *'significantly'* without statistical significance testing, we acknowledge the concern and recognize the importance of substantiating such language. Going forward, we will either rephrase our sentence or ensure its validation through statistical analysis in the final revisions.
> > >
> > > Your insights have contributed to enhancing our work, and we remain dedicated to further refining it based on your valuable input. However, we would appreciate additional clarification regarding the decision to maintain the same rating despite addressing primary issues. Understanding your perspective on this matter would be greatly beneficial as we endeavor to elevate the overall quality of our work.

---

### Official Review · Reviewer_Pq8L · 2023-11-01

**Soundness:** 2 fair
**Presentation:** 3 good
**Contribution:** 2 fair
**Rating:** 5
**Confidence:** 4

**Summary:**

This paper studies training deep neural networks on datasets with a small number of training examples. This paper proposes a new training algorithm that selects subsets of parameters to reinitialize during the training process. For the selection of the parameters, this paper designs a method to generate parameter masks by measuring the influence of masking out one parameter on the loss function and then choosing the top-k parameters with the highest influence values. Experiments are conducted on image classification datasets, including Flower, CUB-200-2011, Stanford Dogs, and FGVC-Aircraft, with ResNet models. The proposed algorithm shows 4% average improvement over previous iterative training and reinitialization algorithms. The baselines include Dense-Sparse-Dense Networks, Born Again Networks, and Knowledge Evolution.  Furthermore, the proposed algorithms are applied to one dataset with corrupted images, CIFAR-10-C, showing consistent improvement over previous approaches. Ablation studies of masking percentages and importance metrics are conducted to confirm the benefits of the algorithm.

**Strengths:**

- Based on the previous knowledge evolution algorithm, this paper proposes to dynamically update the parameter masks through estimating the parameter importance on the downstream datasets.
- This paper provides empirical studies that show the advantage of the proposed algorithm over previous iterative retraining and reinitialization algorithms.

**Weaknesses:**

- Further discussion of the proposed algorithm is needed. For example, how are scores within the SNIP method computed? What is the computation complexity of estimating such scores during the training process? Would it lead to additional overhead?
- More recent baselines need to be compared. This paper conducts a comparison with previous retraining and reinitialization algorithms. However, for training on small datasets, many regularization and training algorithms are proposed, such as sharpness-aware minimization and distance-based regularization. How would the proposed method compare to such methods? Moreover, how does the method perform on transformer-based architectures?
- Further experiments can be conducted to analyze the algorithm. For example, how can one set the number of generations and the masking ratios in the algorithm? How would these parameters affect the model performance? With such a retraining algorithm, would the model converge faster than vanilla fine-tuning?

**Questions:**

See the weaknesses section for the questions.

---

> ### Author Response · Authors · 2023-11-22
> **Reply to Reviewer Pq8L (1/2)**
>
> > Further discussion of the proposed algorithm is needed. For example, how are scores within the SNIP method computed? What is the computation complexity of estimating such scores during the training process? Would it lead to additional overhead?
>
> Thank you for your insightful comments. We appreciate the opportunity to address your concerns regarding the computational aspects of our proposed Selective Knowledge Evolution (SKE) algorithm.
>
> The SNIP method, employed in our proposed Selective Knowledge Evolution (SKE) algorithm, identifies prunable weights at initialization without the need for full network training. It assesses the relevance of each weight through the normalized gradient of the loss with respect to an implicit multiplicative factor on the weights, referred to as "sensitivity." We rank the weights based on their sensitivity and only keep the top parameters based on the sparsity constraint (k), randomly reinitializing the rest before the start of each generation. Further details on this process can be found in Section 3.2.
>
> In our experiments, we consistently maintained a fixed training duration of 200 epochs for each generation, with the number of generations set at 10 for fair comparison. The computational cost of evolutionary training methods, including both KE and SKE, scales linearly with the number of generations (T). For example, if KE is trained for 5 generations, the total computational cost becomes 5T times that of training a single generation. Similarly to ensure fairer comparison, we train a long baseline for the same number of epochs.
> The additional computational cost incurred by SKE for computing data-aware dynamic masking with SNIP is minimal. For instance, on the CUB dataset with a 20% subset, it amounts to 20.3 seconds per generation. This is a one-time calculation performed at the end of each generation. This computational cost can be further reduced by using just 128 samples to estimate the importance without affecting the final performance. Notably, SKE's performance exhibits minimal sensitivity to changes in the subset size, as demonstrated in Appendix, Table 8.
>
> Overall, SKE maintains an equivalent computational cost compared to our long baselines and KE. The slight increase in computational cost in SKE, attributed to weight reinitialization and computing connection sensitivity after each generation, is justified by the substantial improvement in generalization performance. We believe this detailed clarification adequately addresses concerns about computational costs satisfactorily.
>
> > More recent baselines need to be compared. This paper conducts a comparison with previous retraining and reinitialization algorithms. However, for training on small datasets, many regularization and training algorithms are proposed, such as sharpness-aware minimization and distance-based regularization. How would the proposed method compare to such methods?
>
> Thank you for your suggestions. Both methodologies, Selective Knowledge Evolution (SKE) and Sharpness-Aware Minimization (SAM), share the common objective of enhancing generalization. However, they employ distinct strategies to achieve this goal.
> In SAM, the emphasis lies on simultaneously minimizing both loss value and loss sharpness. This involves encouraging parameters that lead to neighborhoods with uniformly low loss, ultimately aiming to improve overall model quality and generalization.
> Contrastingly, SKE follows a unique path. It identifies the least important parameters affecting the loss through data-aware dynamic masking using SNIP. By reinitializing these parameters and subsequently retraining the model, SKE introduces a level of noise or randomness into the network. The alteration of connection sensitivity aims to enhance generalization by freeing up the network's capacity to learn generalized information.
>
> In addition, there are distance-based regularization techniques like DELTA, which proposes a regularized transfer learning framework preserving the outer layer outputs of the target network. These methods align outer layer outputs using attention mechanisms or by regularizing weights with Euclidean distance. Although these approaches effectively address overfitting, they differ from SKE.
> It's important to note that our work stands apart in that it doesn't rely on a teacher network pretrained on a large dataset. Our models are trained from scratch directly on smaller datasets, aiming to avoid overfitting. Conducting direct comparisons with distance-based methods from transfer learning or SAM isn't straightforward and may not align with the specific focus of our paper. These divergent approaches underscore the diversity of techniques employed to tackle the challenge of model generalization.

---

> ### Author Response · Authors · 2023-11-22
> **Reply to Reviewer Pq8L (2/2)**
>
> > Moreover, how does the method perform on transformer-based architectures?
>
> In our work, we have primarily concentrated on convolutional neural network architectures to investigate the efficacy of our proposed Selective Knowledge Evolution (SKE) method. We agree that extending our evaluation to transformer-based architectures is an excellent avenue for future exploration. Transformer architectures, especially in the context of small datasets, present a distinct set of challenges and opportunities, and we believe that investigating the performance of SKE on such architectures could contribute valuable insights.
>
> > Further experiments can be conducted to analyze the algorithm. For example, how can one set the number of generations and the masking ratios in the algorithm? How would these parameters affect the model performance? With such a retraining algorithm, would the model converge faster than vanilla fine-tuning?
>
> We appreciate the insightful suggestion regarding further experiments to analyze our algorithm. Parameters such as the number of generations, sparsity constraint (k), and the percentage of data used to calculate importance play crucial roles in shaping the performance of the retraining algorithm. In practice, the choice of the number of generations is often influenced by the characteristics of the dataset, the complexity of the task, and the availability of computational resources.
>
> To study the impact of masking ratios, we performed experiments by varying the masking ratio K and evaluate the impact on performance.  Table below (Table 4 in the revised paper) shows the effect of varying the number of reinitialized parameters on the performance and generalization of the model. We train the model in evolutionary settings using the SKE framework by varying different percentages of reinitialized parameters (5%, 10%, 20%, 30%, and 40%). Experiments were carried out with ResNet18 on two datasets. The results show that the reinitialization of a 5% percentage of parameters has no impact on performance, while the reinitialization of 30% and more has less impact on test accuracy. We find that reinitialization 20% of the parameters results in the best performance.
>
> |                    | Reinitialized Params (%) | Aircraft | CUB    |
> |--------------------|---------------------------|----------|--------|
> | **SKE**            | 5                         | 65.34    | 69.95  |
> |                    | 10                        | 66.10    | 70.15  |
> |                    | 20                        | **66.63**| **71.37** |
> |                    | 30                        | 64.13    | 68.42  |
> |                    | 40                        | 62.79    | 66.87  |
>
>
> Further, we vary the quantity of data used for importance estimation to analyze its effect on performance. In our experiments, we randomly sampled 20% of the dataset to estimate the importance of the parameters after the end of each generation. Here, we analyze the impact of the number of data points used to determine the important estimation on the final performance. Similarly, we used as few as 128 samples to estimate the important parameters using SNIP. Table 8 in Appendix shows that SKE is not sensitive to the variation in the input data used to estimate the importance, as the final performance remains unchanged.
>
> Our experimental results in Appendix Section A3 indeed reveal that the Selective Knowledge Evolution (SKE) algorithm tends to converge faster than vanilla fine-tuning across various datasets and architectures. The faster convergence observed in SKE can be attributed to its unique methodology of selectively reinitializing less important parameters. By dynamically identifying and retraining these parameters, SKE facilitates faster adaptation to the underlying data distribution, thus contributing to a more efficient learning process.

---

### Meta-Review · Area_Chair_tiZk · 2023-12-10

**Metareview:**

In many practical areas such as medical applications, access to large amounts of training data is often limited, requiring new techniques about how to efficiently train a deep neural network on a small dataset while enhancing its generalization ability. Recent studies have investigated evolutionary or iterative training methods, which involve reinitializing a portion of the network's parameters to boost generalization on small datasets. Although these approaches are beneficial, they typically select parameter subsets randomly and use a fixed mask throughout the training process, which may be suboptimal. This paper draws inspiration by the concept of neurogenesis in the human brain, and proposes an iterative training framework called Selective Knowledge Evolution (SKE). The authors claim that SKE has a dynamic masking strategy that is data-aware and can selectively eliminate redundant connections by assessing their importance. The authors claim that this process allows for the increase of the model's learning capacity through random weight reinitialization. The authors also conduct experiments that suggest that this approach surpasses existing methods in terms of accuracy and robustness.

The reviewers thought the dynamic strategy and the experiments were for the most part interesting. However they thought that further discussion and empirical analysis of the algorithm is needed and that the authors should compare with more recent baselines. They also thought that a lot of experimental details are missing and had concerns about the focus on imbalanced data. The authors did put a lot of effort in addressing the reviewers concerns and I think it did alleviate some of the issues. However, it is clear that a substantial rewriting of the paper is still necessary to fully address the comments. Therefore I can not recommend acceptance at this stage. I encourage resubmission to a future ML venue.

**Justification For Why Not Higher Score:**

The paper IMO is not ready there are many missing details and analysis

**Justification For Why Not Lower Score:**

N/A

---

### Decision · Program_Chairs · 2024-01-16

Reject